# Flavonoid and Phenolic Quantification from Açaí (*Euterpe oleracea* Mart and *Euterpe precatoria* Mart), Mirití (*Mauritia flexuosa* L.), and Cupuassu (*Theobroma grandiflorum* (Wild. Ex Spreng.) Schum) from Vaupés, Colombia, Using LC-QqQ-MS

**DOI:** 10.3390/plants14172632

**Published:** 2025-08-24

**Authors:** Manuel Salvador Rodríguez, Aida Juliana Martínez León, Lina Sabrina Porras, Iván Alejandro Giraldo, Esmeralda Rojas, Fredy Eduardo Lavao, Kaoma Martínez

**Affiliations:** 1Departamento de Ingeniería Civil y Ambiental, Universidad de los Andes, Bogotá 111711, Colombia; manuel-r@uniandes.edu.co (M.S.R.); lporras@uniandes.edu.co (L.S.P.);; 2Gobernación del Vaupés, Departamento Administrativo de Planeación, Mitú 970001, Vaupés, Colombia; 3Gobernación del Vaupés, Secretaria de Agricultura y Desarrollo Productivo, Mitú 970001, Vaupés, Colombia

**Keywords:** Amazonian fruits, flavonoids, phenolic acids, Colombian biodiversity

## Abstract

Amazonian fruits are valued for their rich phytochemical composition, yet limited data exist for species in Colombia. This study aimed to characterize the flavonoid and phenolic acid profiles of *Euterpe olereacea* and *Euterpe precatoria* (açaí), *Mauritia flexuosa* (mirití), and *Theobroma grandiflorum* (cupuassu) from Vaupes, Colombia. Liquid chromatography coupled with triple quadrupole mass spectrometry (LC-QqQ-MS) and LC-QTOF-MS were used to identify and quantify bioactive compounds in fruit pulp samples. A total of 14 flavonoids and 23 phenolic acid derivatives were detected. *M. flexuosa* exhibited the highest total flavonoid content, particularly for catechin (4.86 µg/g). *E. oleracea* and *E. precatoria* showed the highest phenolic acid concentrations, with 4-hydroxybenzoic acid and ferulic acid as dominant compounds. The presence of cyanidin-*O*-glucoside was confirmed in *Euterpe* species but not in *M. flexuosa* or *T. grandiflorum*. Compared to international reports, the Colombian samples generally presented lower concentrations, likely due to genotypic, environmental, and methodological differences. These findings contribute to the phytochemical profiling of underrepresented Amazonian fruits and support their potential for functional food and nutraceutical applications. Further studies are recommended to evaluate the bioavailability and health-promoting effects of these compounds.

## 1. Introduction

Amazonian fruits are a rich source of phenolic compounds, particularly flavonoids (e.g., flavanols, flavones, flavanones, and anthocyanins), phenolic acids, and lignans [1]. These bioactive molecules are classified as secondary metabolites, which are synthesized through the pentose phosphate, shikimate, and phenylpropanoid pathways. Numerous studies have demonstrated their therapeutic potential, including antimicrobial and antioxidant properties [2].

Flavonoids comprise a broad class of natural substances characterized by distinct phenolic structures and are abundantly found in fruits, vegetables, grains, roots, teas, and wines. In biological systems, flavonoids fulfill diverse roles across microorganisms, plants, and animals. In flora, they are responsible for pigmentation and aroma, facilitating pollinator attraction and seed dispersion. Moreover, they function as allelopathic agents, defense compounds against pathogens, and detoxifying molecules. Several flavonoids exhibit antibacterial, antifungal, and antiparasitic activities that inhibit the proliferation and spread of infectious agents [3].

Açaí (*Euterpe oleracea* Mart. and *Euterpe precatoria* Mart.), mirití (*Mauritia flexuosa* L.), and cupuassu (*Theobroma grandiflorum* (Wild. ex Spreng.) Schum.) are native fruits of the Amazon basin. Most existing phytochemical characterizations have been conducted on Brazilian specimens [4]. However, considering the vast geographical scope and ecological variability of the Amazon region, including differences in climate, soil, and biodiversity, it is plausible that the same species in other countries may exhibit distinct profiles and concentrations of flavonoids and phenolic constituents [5].

The department of Vaupés, located in the Colombian Amazon, spans over 54,000 km^2^ and is renowned for its exceptional biological diversity and favorable environmental conditions, which support the growth of both native and introduced plant species. For centuries, indigenous communities in Vaupés have utilized açaí, mirití, and cupuassu as traditional sources of nutrition and medicine. Despite this longstanding cultural significance, scientific studies on the biochemical composition of these fruits in Colombia remain limited, underscoring the need for further research and dissemination [6].

Açaí (*E. oleracea* and *E. precatoria*) is a palm from the *Arecaceae* family native to lowland tropical regions of South America. *Euterpe* species are differentiated based on the number of stems, flower traits, trichomes, and eophylls. *E. oleracea* is a multi-stemmed palm, while *E. precatoria* is a solitary palm [7]. Its fruits contain notable levels of macronutrients (carbohydrates, proteins, dietary fiber, and lipids rich in mono- and polyunsaturated fatty acids), as well as minerals such as phosphorus (P), manganese (Mn), iron (Fe), and zinc (Zn). In addition, they are rich in bioactive compounds, including anthocyanins, proanthocyanidins, flavonoids, phenolic acids, and stilbenes such as resveratrol [5]. Several studies report therapeutic activities associated with açaí’s polyphenolic content, including antiproliferative effects on HT-29 colon cancer cells, hepatoprotection against steatosis, antiplasmodial action, neuroprotective mechanisms, and anti-leukemic activity [2].

Cupuassu (*T. grandiflorum*), an Amazonian tree crop, belongs to the *Malvaceae* family. It is endemic to the Brazilian rainforests and has been successfully introduced into Colombia’s humid tropics through the Putumayo River from Manaus [8]. The fruit emits a distinctive aroma due to its volatile ester compounds (e.g., ethyl acetate, ethyl butanoate, ethyl propanoate, and ethyl hexanoate). Its pulp is nutritionally rich, containing carbohydrates—predominantly sucrose—and a high concentration of fatty acids such as palmitic, linoleic, and α-linolenic acids. It also provides essential micronutrients, including potassium (K), magnesium (Mg), and phosphorus (P) [6]. Notably, cupuassu demonstrates significant antioxidant capacity due to its elevated content of ascorbic acid and flavonoids, mainly catechin, epicatechin, quercetin, and kaempferol, among others [9].

Mirití (*M. flexuosa*) is an endemic palm of the Amazon, belonging to the *Arecaceae* family, which is widely distributed across South America. Its fruits are nutritionally dense and recognized for their abundance of bioactive molecules. They contain high levels of lipids, proteins, fiber, tannins, phenolics, flavonoids, copper, and potassium. Mirití is also distinguished by its high total carotenoid content, which contributes to antioxidant potential and serves as a precursor of vitamin A. Empirical studies have demonstrated a robust positive correlation between total phenolic content and antioxidant activity in this species [5,10,11].

In recent years, the scientific community has shown growing interest in the characterization of secondary metabolites due to their potential applications across food, pharmaceutical, and cosmetic industries. While general information on the phytochemical content of Amazonian fruits is available, there is a lack of region-specific data, particularly for Vaupés, Colombia, that accounts for environmental and genetic variability affecting metabolite composition.

In this context, the present study aims to characterize the flavonoid and phenolic profiles of three Amazonian fruits—açaí, mirití, and cupuassu—collected in Vaupés, using liquid chromatography coupled with triple quadrupole mass spectrometry (LC-QqQ-MS). This technique offers high sensitivity, selectivity, and quantification accuracy for targeted metabolite analysis. By employing LC-QqQ-MS, we aim to obtain precise and reproducible quantitative data. This method provides a comprehensive phytochemical assessment of key Amazonian fruits from an underexplored region, contributing valuable new data to both biodiversity research and functional food science.

## 2. Results and Discussion

### 2.1. Flavonoid Contents

The flavonoid and phenolic acid profiles of *Euterpe oleracea*, *Mauritia flexuosa*, and *Theobroma grandiflorum* were analyzed using liquid chromatography coupled with triple quadrupole mass spectrometry (LC-QqQ-MS). The figures in the Appendix A show the chromatograms of phenolic and flavonoid compounds (Appendix A). Table 1 summarizes the flavonoid concentrations quantified in the fruit samples. Among the three species, *M. flexuosa* exhibited the highest total flavonoid content. Notably, the flavonoid content was greater for Euterpe species (16.23 µg/g DW), followed by *M. flexuosa* (10.61 µg/g DW) and *T. grandiflorum* (6.43 µg/g DW). At the individual sample level, the highest compound-specific concentrations observed were rutin in açaí (4.4 µg/g DW), epicatechin in cupuassu (5.76 µg/g DW), and catechin in mirití (4.86 µg/g DW).

These findings align with previous phytochemical reports for *E. oleracea* and *E. precatoria*, which have identified flavonoids as dominant secondary metabolites. Dantas et al. [12] documented concentrations of anthocyanins reaching 198.98 mg/100 g (DW), flavanols (including catechin and epicatechin) at 50.65 mg/100 g DW, and flavonols—such as quercetin-3-glucoside, rutin, kaempferol-glucoside, and naringenin—at 10.88 mg/100 g DW.

In the case of *M. flexuosa* and *T. grandiflorum*, kaempferol concentrations observed here agree with earlier reports by Carmona-Hernández et al. [13]. These findings highlight the value of LC-QqQ-MS in Amazonian fruit characterization [14,15].

A total of 14 distinct flavonoids were identified in *Euterpe oleracea* and *Euterpe precatoria* (açaí) samples. In *E. oleracea*, rutin was the predominant compound (4.44 µg/g DW), followed by catechin (1.35 µg/g DW), luteolin (0.97 µg/g DW), and diosmetin (0.72 µg/g DW). Conversely, in *E. precatoria*, catechin was the most abundant (3.39 µg/g DW), followed by luteolin (1.23 µg/g DW), rutin (0.91 µg/g DW), and taxifolin (0.59 µg/g DW), while the remaining compounds were present in concentrations below 0.35 µg/g DW.

Notably, cyanidin, a key anthocyanin in açai, was not detected using the targeted method due to the absence of a specific analytical standard. To address this limitation, an untargeted metabolomic analysis was conducted (Table 2), revealing the presence of cyanidin-*O*-glucoside (12.46% in EO and 10.92% in EP). Previous studies [16] have reported the presence of cyanidin-3-rutinoside and cyanidin-3-glucoside in açai, which is consistent with our findings in the analyzed samples.

Furthermore, the untargeted metabolomic approach identified different flavonoids across the samples. Specifically, 20 flavonoids, including kaempferol-*O*-rutinoside, fisetin, procyanidin B2, and rutin, among others, were detected in *Euterpe* species, followed by 12 flavonoid compounds in *M. flexuosa* and 4 in *T. grandiflorum*. Interestingly, the targeted LC-QqQ-MS method detected a greater number of flavonoids in *M. flexuosa* and *T. grandiflorum* compared to the untargeted approach. This observation suggests that the targeted method may offer greater sensitivity for flavonoid detection in certain species [17].

These findings highlight the interspecific variation in flavonoid accumulation within the *Euterpe* genus. The *E. oleracea* species is native to the central Amazon region and typically grows in multi-stemmed clusters, while the *E. precatoria* genus is dominant in the Amazon biome and is found as a solitary palm in areas of permanent flooding [7].

Although previous studies on commercial açaí products, such as those by Costa et al. [18], have reported higher total flavonoid concentrations, these differences may reflect the variation in genotypes, environmental factors, or sample processing methods. Interestingly, the concentrations of catechin and epicatechin detected in our study exceeded those reported by Garzón et al. [19]. These catechins are well recognized for their antioxidant, antimicrobial, anti-inflammatory, and antiproliferative activities [18,19,20], and their elevated presence in *Euterpe* species may have functional relevance for nutritional and pharmacological applications.

In *Mauritia flexuosa* (mirití), 13 flavonoids were identified. Catechin was the predominant compound with 4.86 µg/g DW, followed by luteolin with 1.72 µg/g DW and rutin with 1.22 µg/g DW. Other compounds like (+)-taxifolin and (+/−)-naringenin were present in concentrations below 0.82 µg/g DW. Although previous studies have reported significantly higher catechin and luteolin concentrations in *M. flexuosa*, such as those by Bataglion et al. [14], our values are more consistent with the levels observed by Tauchen et al. [15] in mesocarp tissues. The variability across studies may be due to differences in tissue type, ripeness stage, and extraction methodologies, as noted by do Nascimento et al. [21]. Additionally, environmental factors such as photoperiod, rainfall, and soil conditions may influence secondary metabolite profiles [11]. In the untargeted metabolomic analysis, cyanidin-O-galactoside was not detected in *M. flexuosa* samples. This finding is consistent with previous reports [5] that identified cyanidin-3-rutinoside and cyanidin-3-glucoside as the principal anthocyanins.

In *Theobroma grandiflorum* (cupuassu), 12 flavonoids were detected. Epicatechin was by far the most abundant (5.75 µg/g DW), followed by catechin (0.56 µg/g DW), while the remaining compounds were below 0.08 µg/g DW. Prior research has reported significantly higher values: catechin at 2.20 mg/g DW and epicatechin at 60.40 mg/g DW. Cuéllar Álvarez et al. [22] similarly found elevated concentrations in cupuassu beans—catechin (10.06 ± 20.11 mg/g) and epicatechin (5.74 ± 5.83 mg/g). These pronounced differences can likely be attributed to variations in extraction techniques. For example, Benlloch-Tinoco et al. [23] demonstrated that using 12% ethanol in the extraction solvent maximizes flavonoid recovery from dehydrated pulp. In contrast, the present study employed 80% ethanol, which may have altered the solvent polarity and thereby reduced the extraction efficiency of certain hydrophilic flavonoids. In the untargeted metabolomic analysis, cyanidin-O-galactoside was not detected in *T. grandiflorum* samples, which is consistent with previous research reporting cyanidin-3-rutinoside and cyanidin-3-glucoside as the principal anthocyanins in cupuassu [13].

### 2.2. Phenolic Acid Content

The types and concentrations of phenolic acids identified in the analyzed fruit samples are shown in Table 3. It can be observed that *E. oleracea* and *E. precatoria* exhibited higher values of phenolic acids (612.83 μg/g DW and 422.35 μg/g DW) followed by *M. flexuosa* (577.02 μg/g DW). *T. grandiflorum* exhibited a lesser value of 17.37 μg/g DW. The mirití sample showed the highest concentration of 4-hydroxybenzoic acid with 317.8 µg/g, followed by açaí sample species *E. oleracea* with 254.3 µg/g and açaí sample species *E. precatoria* with 237.0 µg/g. Previous studies have also reported the presence of p-hydroxybenzoic acid, ferulic acid, vanillic acid, and syringic acid in both açaí species fruits [24]. For mirití mesocarp samples, Tauchen et al. [15] reported the presence of ferulic acid, vanillic acid, caffeic acid, and p-Coumaric acid, among others. Marty et al. [25] reported the presence of 4-hydroxybenzoic acid and ferulic acid in cupuassu pulp.

Phenolic acids have been associated with antioxidants, as well as anti-inflammatory, antimicrobial, and metabolic process-modulating processes [26].

For açaí, a total of 23 phenolic acid derivatives were identified. Regarding phenolic compounds, the results for açaí indicate that for the species *E. oleracea*, the concentration of most compounds is higher than that found in samples of the species *E. precatoria*. The concentration of 4-hydroxybenzoic acid for the species *E. oleracea* was 254.3 µg/g DW, while for the species *E. precatoria*, it was 237.0 µg/g DW. In the case of 3,5-dihydroxybenzoic acid, the concentration for *E. oleracea* was higher, with 121.5 µg/g DW, and for *E. precatoria*, it was 40.1 µg/g DW. In addition, the concentration of ferulic acid for *E. oleracea* was 121.52 µg/g DW, and for *E. precatoria*, it was 70.48 µg/g DW. Compared to previous studies, the concentrations of p-hydroxybenzoic acid and ferulic acid observed in the present study were higher than those reported by [24], who documented levels of 1.80 ± 0.13 mg/kg of p-hydroxybenzoic acid and 0.98 ± 0.10 mg/kg of ferulic acid.

A total of 22 phenolic acid derivatives were identified in the *M. flexuosa* samples. The results for mirití indicate that 4-hydroxybenzoic acid was the compound with the highest concentration, 317.83 µg/g DW, followed by ferulic acid with 98.98 µg/g DW, 3,5-dihydroxybenzoic acid with 56.25 µg/g DW, and vanillic acid with 45.19 µg/g DW. Compared to previous studies, the concentration of ferulic acid was higher than that reported by the authors of [15], who detected levels of 93.4 ± 0.13 ng/g DW of ferulic acid in the mesocarp of miriti. It is also notable that the authors of [15] did not report the presence of 4-hydroxybenzoic acid in their analysis.

In the case of cupuassu, 19 phenolic derivatives were detected. The results of the phenolic compounds for cupuassu indicate that terephthalic acid was the compound with the highest concentration, 12.5 µg/g, followed by 3,5-dihydrobenzoic acid with 1.25 µg/g. The remaining compounds were present at concentrations below 0.68 µg/g. In contrast, in a study by Tauchen et al. [15] in cupuassu pericarp, neither terephthalic acid nor 3,5-dihydroxybenzoic acid was detected. Instead, they reported the presence of other phenolic acids, including ferulic acid (76.8 ± 0.2 ng/g dw), gallic acid (6.7 ± 0.2 ng/g), salicylic acid (121.6 ± 0.4 ng/g), and syringic acid (497.5 ± 0.7 ng/g), which were detected at lower concentrations in the present study. Similarly, Marty et al. [25] did not detect terephthalic acid or 3,5-dihydroxybenzoic acid, but they identified the presence of 4-hydroxybenzoic acid, ferulic acid, gallic acid, caffeic acid, and syringic acid.

Although this study did not include a genetic or phylogenomic analysis, the phytochemical differences observed among *E. oleracea*, *E. precatoria*, *M. flexuosa*, and *T. grandiflorum* may, in part, reflect their evolutionary divergence. Research shows that the evolution of secondary metabolite biosynthetic pathways, like flavonoid production, has deep evolutionary roots across plant lineages [27]. In our case, the distinct flavonoid, polyphenol, and phenolic acid ratios found in *M. flexuosa* versus *T. grandiflorum* support this pattern [13], while the abundant phenolic and flavonoid composition shared by the Euterpe species is consistent with conserved biosynthetic pathways between the *E. oleracea* and *E. precatoria* sister species [28].

## 3. Materials and Methods

### 3.1. Reagents and Chemicals

The commercial standards for the identification and relative quantification of phenolic and flavonoid compounds were obtained from MetaSci^®^ (Toronto, ON, Canada). A standard solution with a concentration of 100 ppb was used for this purpose. A table in the Appendix A shows the standards used in this study (Appendix A).

### 3.2. Fruit Collection and Sample Preparation

Ripe açaí (*E. oleracea* and *E. precatoria*), mirití (*M. flexuosa*), and cupuassu (*T. grandiflorum*) were collected in Mitú, Vaupes, Colombia (latitude: 1°14’54.36″ N; longitude: 70°14’24.66″ W) between February and May. Selected fruits were washed and disinfected, and then immersed in water at ambient temperature for 12 h to facilitate pulp detachment. Pulp extraction was carried out using a mechanical pulper that separated seeds from the mesocarp. The resulting pulp was transferred into Falcon tubes and stored at −80 °C for 24 h prior to freeze-drying. For *T. grandiflorum*, pulp was manually extracted by separating it from the woody exocarp. Subsequently, it was stored in plastic bags at −80 °C for 24 h and subjected to freeze-drying. The freeze-drying process was conducted using a Labconco, 7400040, freeze-dryer (Labconco Corporation, Kansas City, MO, USA). The freeze-drying conditions included −80 °C and 0.140 mbar of vacuum for 20 min, followed by −55 °C and 0.140 mbar of vacuum for 24 h. Only the mature stages were considered for this analysis. The study was approved by the “Collection Framework” granted to the Universidad de los Andes by Resolution No. 002377, 2024-RCM0014-00-2024, and Addendum No. 2 of the Framework Contract for Access to Genetic Resources and Derived Products No. 288, 2020, file RGE338-2. Post-harvested fruits were placed in plastic bags and transported to the laboratory.

### 3.3. Extraction Procedures

Two sequential extractions were performed on the same freeze-dried sample to obtain a comprehensive profile of phenolic compounds: (i) a hydroalcoholic extraction using 20 to 25 mg of lyophilized sample in 1 mL of 80% ethanol targeting free phenolics and flavonoids and (ii) a sequential alkaline–acid hydrolysis to release bound phenolic compounds. The samples were shaken at 200 rpm for 10 min at 25 °C and then centrifuged at 5000× *g* for 10 min at the same temperature. The remaining pellet was subjected to alkaline hydrolysis by adding 600 µL of 4 M NaOH and ultrasound treatment for 90 min at 40 °C. Subsequently, acid hydrolysis was performed by adjusting the pH to approximately 2 with concentrated HCl, followed by centrifugation at 2000× *g* for 5 min at 25 °C. Then, 1 mL of ethyl acetate was added to the supernatant and centrifuged again. Extracts of phenolic compounds, both free and bound, were evaporated in SpeedVac and resuspended in 500 µL of a mixture of methanol, acetonitrile, and Milli-Q water (2:5:93, *v*/*v*) [29].

### 3.4. Phenolic and Flavonoid Compound Profile by Liquid Chromatography Coupled to Mass Spectrometry with a Triple Quadrupole Analyzer (LC-QqQ-MS)

An analysis of free and bound phenolic and flavonoid compounds in ripe freeze-dried samples of açaí, mirití, and cupuassu was carried out using an Agilent Technologies 1260^®^ Agilent Technologies Inc., Santa Clara, CA, USA, liquid chromatographer coupled to a 6470 triple quadrupole mass analyzer with electrospray ionization^®^ Agilent Technologies Inc., Santa Clara, CA, USA. The methodology proposed by [29] was used and adjusted for this analysis. A total of 3 μL of the sample was injected into a C18 column (InfinityLab Poroshell 120 EC-C18, 2.1 × 150 mm, 2.7 μm, Agilent Technologies Inc., Santa Clara, CA, USA) at 30 °C, using a gradient elution composed of mobile phase A (0.1% *v*/*v* formic acid in Milli-Q water) and mobile phase B (0.1% *v*/*v* acetonitrile) at a constant flow rate of 0.4 mL/min. The chromatographic gradient began with 20% of phase B and was held constant for the first 6 min. It was then gradually increased to reach 80% of phase B by minute 16. At that point, the conditions were maintained for an additional 4 min. Subsequently, the gradient returned to the initial conditions, with a re-equilibration period of 5 min. Mass spectrometry detection was performed in MRM (multiple reaction monitoring) mode at 3000 V, using an ESI source in negative ionization mode. Nitrogen was used as the nebulizing gas at 50 psi, with a drying temperature of 325 °C and a flow rate of 8 L/min. The sheath gas temperature was 350 °C with a flow rate of 11 L/min. The collision gas used was nitrogen (99.999% purity). The programs MassHunter Acquisition (B.10.0.127), Qualitative (B.10.0.1035.0), and Quantitative (B.10.0.707.0) were used for MRM profiling. The specific MRM transitions (precursor and product ions), along with the fragmentation voltages and retention times for each analyte, are presented in Appendix A.

### 3.5. Anthocyanin Profile via Untargeted Metabolomic LC-QTOF-MS

The anthocyanin profile was determined using 5 mg of each freeze-dried sample mixed with 200 µL of MeOH. The samples were mixed by vortex for 15 min, followed by 10 min in ultrasound. The samples were centrifuged at 1500 rpm for 10 min. The supernatant was filtered with PTFE membranes with a diameter of 0.22 µm. Liquid chromatography was performed using an Agilent Technologies 1260 Infinity system, Agilent Technologies Inc., Santa Clara, CA, USA, coupled to a quadrupole time-of-flight (Q-TOF) 6545 mass spectrometer with electrospray ionization (ESI). A 1 µL aliquot of each sample was injected into an InfinityLab Poroshell 120 EC-C18 column (3.0 × 100 mm, 2.7 µm. Agilent Technologies Inc, Santa Clara, CA, USA) maintained at 30 °C. The chromatographic separation was carried out using a gradient elution consisting of 0.1% (*v*/*v*) formic acid in Milli-Q water (mobile phase A) and 0.1% (*v*/*v*) formic acid in acetonitrile (mobile phase B), at a constant flow rate of 0.4 mL/min. Mass spectrometric detection was performed in both positive and negative ESI modes using full scan acquisition from 50 to 1100 *m*/*z* and MS/MS from 100 to 1700 *m*/*z*. For accurate mass calibration throughout the analysis, the following reference ions were used: *m*/*z* 121.0509 (C_5_H_4_N_4_) and *m*/*z* 922.0098 (C_18_H_18_O_6_N_3_P_3_F_24_) in positive mode, and *m*/*z* 112.9856 [C_2_O_2_F_3_(NH_4_)] and *m*/*z* 1033.9881 (C_18_H_18_O_6_N_3_P_3_F_24_) in negative mode. The Q-TOF instrument was operated in a 4 GHz high-resolution mode. The chromatographic gradient began with 30% of mobile phase B and was held for 15 min; then, it was increased to 98% B over 2 min and maintained at this composition until 19 min. The system was then returned to the initial conditions over 1 min, followed by a 5 min re-equilibration period.

### 3.6. Data Treatment and Metabolite Annotation

LC-MS data processing was performed with the Agilent Mass Hunter Profinder 10.0 software, using the Recursive Molecular Extraction algorithm. The annotation of flavonoid metabolites was carried out using different tools, such as exact mass, molecular formula, mass spectra, and retention times. Each metabolite is reported with the confidence level ID according to the guidelines established by the Metabolomics Standards Initiative (MSI). To search for metabolites in the different databases of the CEU MASS MEDIATOR, a search was carried out in batch. To confirm the annotation of metabolites, MS-DIAL software- Version MS-DIAL 4.90 was used, as well as the following libraries from the MONA MassBank of North America: MoNA—export-LC-MS-MS_Positive_Mode, MoNA—export-MassBank, and MoNA—export-Vaniya-Fiehn_Natural_Products library. GC-MS data processing was performed with the Agilent Mass Hunter Unknown Analysis B.10.00, and the identification was performed by searching two specific libraries: Fiehn GC-MS Metabolomics RTL Library version 2013 and NIST Mass Spectral Reference Library (National Institute of Standards and Technology) version 2017.

## 4. Conclusions

The flavonoid and phenolic acid content in lyophilized pulp extracts of *E. oleracea, E. precatoria, M. flexuosa*, and *T. grandiflorum* samples from Vaupes, Colombia, using LC-Qq-MS was evaluated. The results suggest that the content of phenolic acids was higher than the flavonoid content, and that the açaí and miriti samples exhibited greater values of flavonoids and phenolic acids compared to cupuassu samples. *M. flexuosa* exhibited the highest total flavonoid content, with catechin, epicatechin, and rutin as the most abundant compounds. Phenolic acid analysis revealed that *E. oleracea* and *E. precatoria* contained the highest concentrations, with most of the related compounds found to have antioxidant, anti-inflammatory, and antimicrobial properties. When compared to international data, the flavonoid and phenolic acid concentrations observed in Colombian samples were generally lower than those reported in Brazilian studies. These discrepancies may be attributed to differences in extraction protocols, environmental conditions, fruit maturity, and genetic variability. Despite these differences, the Colombian fruits demonstrated a rich diversity of flavonoids and phenolic acids, including compounds with antioxidant, anti-inflammatory, and antimicrobial properties. These results represent a contribution to scientific knowledge of bioactive compounds in Colombia and the Amazon region. Future research should focus on evaluating the anthocyanin profile and bioavailability and exploring the health benefits of these compounds in clinical settings.

## Figures and Tables

**Table 1 plants-14-02632-t001:** Flavonoid composition of three different Amazonian fruits (μg/g fruit sample).

Compound	Transition (*m*/*z*)	RT (min)	EO ^1^	EP ^2^	TG ^3^	MF ^4^
Flavonoids			8.57	7.66	6.43	10.61
(+)-Catechin (Hydrate)	289.0 → 245.0	1.32	1.35	3.39	0.56	4.86
(−)-Epicatechin	289.0 → 109.0	1.50	0.33	0.13	5.75	
Rutin	609.0 → 300.1	1.92	4.44	0.91	0.005	1.22
(+)-Taxifolin	303.0 → 285.0	3.18	0.19	0.59	0.08	0.82
Naringin	579.0 → 271.0	3.75			0.01	nd
Phloridzin	435.0 → 273.0	5.89	0.01	0.002	nd	0.003
Baicalin	444.9 → 269.0	7.79	0.01	0.004	0.01	
Luteolin	284.9 → 133.0	10.12	0.97	1.23	0.003	1.72
Morin	301.0 → 121.0	10.18	0.11	0.07	0.01	0.10
Quercetin	300.9 → 121.0	10.18	0.10	0.06	0.004	0.10
(+/−)-Naringenin	270.9 → 150.9	11.37	0.29	0.34	nd	0.51
Apigenin	268.9 → 151.0	11.45	nd	0.06	nd	0.07
Phloretin	273.0 → 167.0	11.52	0.002	0.003	nd	0.004
Kaempferol	284.9 → 93.1	11.69	0.04	0.52	0.01	0.73
Diosmetin	299.0 → 284.1	11.77	0.72	0.35	0.001	0.47
Biochanin A	282.9 → 268.0	14.13			0.001	

^1^ EO: *E. oleracea*; ^2^ EP: *E. precatoria*; ^3^ TG: *T. grandiflorum*; ^4^ MF: *M. flexuosa*; nd: no detection.

**Table 2 plants-14-02632-t002:** Untargeted metabolomic analysis for flavonoid profile of three different Amazonian fruits (%RA).

Compound	Molecular Formula	Molecular Weight (g/mol)	RT (min)	EO ^1^	EP ^2^	TG ^3^	MF ^4^
Procyanidin B2	C_30_H_26_O_12_	578.1424	7.35	0.21	0.26		0.11
Cyanidin-*O*-glucoside	C_21_H_21_O_11_	449.1084	8.43	12.46	10.92		
Fisetin	C_15_H_10_O_6_	286.0477	8.45	23.42	16.53		
Kaempferol-*O*-rutinoside	C_27_H_30_O_15_	594.1585	8.66	19.30	22.75		
Epigallocatechin gallate	C_22_H_18_O_11_	458.0849	8.76			0.26	
Procyanidin B1	C_30_H_26_O_12_	578.1424	9.03			0.85	
Rutin	C_27_H_30_O_16_	610.1534	9.28	1.96	1.32		
Epicatechin	C_15_H_14_O_6_	290.079	9.68	0.35	0.29		0.01
Taxifolin	C_15_H_12_O_7_	304.0583	10.11	0.49	0.70		
Procyanidin C1	C_45_H_38_O_18_	866.2058	10.23			0.34	
Manghaslin	C_33_H_40_O_20_	756.2113	10.50	0.01	0.02		0.30
Isovitexin	C_21_H_20_O_10_	432.1057	11.94	4.38	1.21		0.04
Hyperoside	C_21_H_20_O_12_	464.0955	12.35	0.45	0.11		0.31
Kaempferol-*O*-glucoside	C_21_H_20_O_11_	448.1006	12.44	3.00	1.44		0.01
Isokaempferide	C_16_H_12_O_6_	300.0634	13.08	0.05	0.02		
Methylquercetin	C_16_H_12_O_7_	316.0583	13.18	0.08	0.03		
Rhamnetin	C_16_H_12_O_7_	316.0583	13.18				0.01
Diosmetin	C_16_H_12_O_6_	300.0634	13.31	0.67			
Naringenin-*O*-glucoside	C_21_H_22_O_10_	434.1213	13.96	0.25	0.71		0.02
Luteolin	C_15_H_10_O_6_	286.0477	17.15	0.61	0.24		0.01
Santin	C_18_H_16_O_7_	344.0896	17.72	0.69			
2′,6′-Dihydroxy-4′-methoxydihydrochalcone	C_16_H_16_O_4_	272.1049	17.93				0.47
Pinocembrin	C_15_H_12_O_4_	256.0736	18.57	0.03	0.03		0.64
3,7-Dihydroxy-5,3′,4′-trimethoxyflavone	C_18_H_16_O_7_	344.0896	18.62	0.40			
Catechin	C_15_H_14_O_6_	290.0790	33.91	1.05	4.00	0.43	0.01

^1^ EO: *E. oleracea*; ^2^ EP: *E. precatoria*; ^3^ TG: *T. grandiflorum*; ^4^ MF: *M. flexuosa*.

**Table 3 plants-14-02632-t003:** Phenolic acid composition of three different Amazonian fruits (μg/g fruit sample).

Compound	Transition (*m*/*z*)	RT (min)	EO ^1^	EP ^2^	TG ^3^	MF ^4^
Phenolic acids			612.83	422.35	17.37	577.02
Gallic acid	169.0 → 125.0	1.01	0.13	0.10	0.07	0.15
3,5-Dihydroxybenzoic acid	152.9 → 108.9	1.20	122.16	40.14	1.25	56.25
Chlorogenic acid	353.0 → 191.0	1.20	1.38	0.05	0.08	0.07
2,3,4-Trihydroxybenzoic acid	168.9 → 150.9	1.29	0.86	0.34	nd	0.47
Dihydrocaffeic acid	180.9 → 136.9	1.56	0.22	0.16	0.13	nd
4-Hydroxybenzoic acid	137.0 → 93.0	1.64	254.34	237.02	0.49	317.83
Terephthalic acid	165.0 → 121.0	1.66	11.24	9.32	12.48	12.73
Caffeic acid	179.0 → 135.0	1.67	4.84	1.07	0.03	1.51
Syringic acid	197.0 → 182.0	1.74	2.90	5.62	0.54	7.89
Gentisic acid	153.0 → 108.0	1.74	0.09	0.10	nd	0.16
4-Acetocatechol	151.0 → 108.0	1.75	1.39	0.33	0.01	0.44
Vanillic acid	167.0 → 151.9	1.78	29.77	32.61	0.68	45.19
2,3-Dihydroxybenzoic acid	153.0 → 109.0	2.05	0.18	0.04	0.01	0.05
2,4-Dihydroxybenzoic Acid	153.0 → 109.0	2.05	0.12	0.08	nd	0.12
p-Coumaric acid	163.0 → 119.0	2.52	53.99	15.68	0.26	22.29
Hydroferulic acid	195.0 → 136.0	2.60	0.32	0.08	0.04	0.15
Sinapic acid	223.0 → 193.0	2.80	6.30	8.60	0.09	12.13
m-Hydrocoumaric acid	165.0 → 121.0	2.90	0.07	0.03	nd	0.04
Ferulic acid	193.0 → 134.0	2.94	121.52	70.48	0.60	98.98
m-Coumaric acid	163.0 → 119.0	3.44	0.12	0.09	0.04	0.11
Acetylphloroglucinol	167.0 → 123.0	3.89	nd	0.10	nd	0.13
trans-2-Hydroxycinnamic acid	163.0 → 119.0	4.80	0.44	0.11	0.43	0.09
Salicylic acid	137.0 → 93.0	5.53	0.43	0.20	0.13	0.23
3,4,5-Trimethoxycinnamic acid	237.0 → 102.9	9.42			nd	nd
Caffeic acid phenethyl ester	283.0 → 135.0	14.12	nd	nd	0.005	

^1^ EO: *E. oleracea*; ^2^ EP: *E. precatoria*; ^3^ TG: *T. grandiflorum*; ^4^ MF: *M. flexuosa*; nd: no detection.

## Data Availability

The data presented in this study are available within the article.

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
