# Peer review of "Flavonoid and Phenolic Quantification from Açaí (Euterpe oleracea Mart and Euterpe precatoria Mart), Mirití (Mauritia flexuosa L.), and Cupuassu (Theobroma grandiflorum (Wild. Ex Spreng.) Schum) from Vaupés, Colombia, Using LC-QqQ-MS"

_plants, 2025, doi:10.3390/plants14172632_

Round 1
Reviewer 1 Report
Comments and Suggestions for Authors
This manuscript describes the characterization and quantification of flavonoid and phenolic acids of three Amazonian fruits: Euterpe olereacea, Euterpe precatoria (açaí) Mauritia flexuosa (mirití) and Theobroma grandiflorum (cupuassu) from Vaupes, Colombia using liquid chromatography coupled with triple quadrupole mass spectrometry (LC-QqQ-MS).
Although there are some previous studies involving the phenolic profile of these fruits, the authors point out the importance of the work, considering the vast geographical scope and ecological variability of the Amazon region, including differences in climate, soil, and biodiversity, been plausible that the same species in other countries may exhibit distinct profiles and concentrations of phenolic constituents.
The article has some merit for publication in Plants, however there are some items that need to be reviewed and/or clarified.
Please consider the comments below:
Results and Discussion
Line 106: Dantas et al. 2019 (?) – The citation is not formatted correctly, and the reference is not listed in the reference list.
Line 125: Costa et al. (2021) - The citation is not formatted correctly.
Materials and Methods
Were all characterized compounds compared to their respective standards analyzed under the same conditions?
In the compound quantification experiment, it wasn't clear exactly how the experiments were conducted. Were the samples injected in replicates? Was there any statistical analysis?
Was the analytical method previously validated?
Author Response
"Please see the attachment"

Reviewer 2 Report
Comments and Suggestions for Authors
In the manuscript by Martínez León et al. entitled »Flavonoid and phenolic quantification from açaí (Euterpe oleracea Mart, Euterpe precatoria Mart), mirití (Mauritia flexuosa L.), and cupuassu (Theobroma grandiflorum (Wild. Ex Spreng.) Schum) from Vaupés - Colombia using LC-QqQ-MS«, the authors analysed the flavonoid and phenolic acid profiles of three Amazonian fruits — Euterpe oleracea, Euterpe precatoria (açaí), Mauritia flexuosa (mirití), and Theobroma grandiflorum (cupuassu) — collected in Vaupés, Colombia. Using LC-QqQ-MS, 14 flavonoids and 23 phenolic acid derivatives were identified. M. flexuosa showed the highest total content of flavonoids, especially catechin in overripe fruits. E. oleracea and E. precatoria had the highest phenolic acid concentrations, with 4-hydroxybenzoic acid and ferulic acid being the dominant compounds. Compared to the Brazilian samples, the Colombian fruits had lower concentrations of compounds, probably due to genetic, environmental and methodological differences. These results contribute to the establishment of a phytochemical profile of the underrepresented Amazonian fruits and support their potential for functional food and nutraceutical applications. Further research is recommended to assess bioavailability and health benefits.
The language in the manuscript is generally clear and easy to read, although there are passages where the language could be improved to increase clarity and readability. Otherwise, the paper is logically organised and the sections are clearly defined.
The work can be considered original, although the originality stems mainly from the focus on the analysed topics (Amazonian fruits from Colombia). With regard to the concept of the study and the analytical approach (determination of flavonoids and phenolic acids by LC-QqQ-MS), it would be pretentious to categorise the study as highly original. Nevertheless, this part is interesting as it allows the simultaneous quantification of 14 flavonoids and 23 phenolic acid derivatives, which provides a good starting point for comparisons within the present study and with results of other similar studies.
Regarding the reliability of the results, the standard analytical approach (which, as mentioned above, is not very original) allows obtaining quite reliable reference data for further studies. On the other hand, it is perhaps a pity that the authors did not try to go much further than just obtaining new data. What I miss in this case is a more in-depth discussion of why the flavonoid and phenolic acid concentrations observed in Colombian samples were generally lower than those reported in Brazilian studies. For example, using the same procedure (an additional experiment), it would be possible to obtain information on how different extraction protocols affect the results, and such information is important when comparing results obtained with different extraction approaches. While I welcome the move to analyse Mauritia flexuosa fruit at two different stages of ripeness, I cannot understand why this part of the study was not extended to two other fruits and/or to more stages of ripeness than two – especially when the authors state that the ripeness of the fruit [line 276] has an impact on the flavonoid and phenolic acid profile.
As for the overall assessment of the article, I would say that establishing reliable flavonoid and phenolic acid profiles of plants for which not much data is available is undeniably its greatest strength, which makes it a good candidate for publication in Plants, especially if the results can lead to nutraceutical and food science applications. However, the study lacks both a deeper analysis of the influence of ripeness stage and the evolutionary relationship on the flavonoid and phenolic acid profile. In addition, the study does not include bioactivity assay studies (e.g. antioxidant capacity, antimicrobial tests), which is also important if applications are sought.
To summarise, I would say that the article could be published in Plants, but before publication I suggest a minor revision:
1.) Expand the part of the article that discusses the influence of ripeness stage on flavonoid and phenolic acid profiles.
2.) The article does not explicitly address the genetic or evolutionary relationships between the species studied (Euterpe oleracea, Euterpe precatoria, Mauritia flexuosa and Theobroma grandiflorum). Although the authors mention that differences in phytochemical composition could be due to genotypic variability, no mention is made of phylogenetic relationships, evolutionary divergence or genetic lineage. It would therefore be interesting to know if there is a correlation between the genetic/evolutionary relationship between these plants and the flavonoid and phenolic acid profile obtained.
3.) The names of some authors of the paper in Ref 20 are not written correctly. Is the name of the journal “Acta Agronómica” written in its abbreviated form (names of journals should be written as abbreviated names in References of MDPI journals)?
Author Response
"Please see the attachment"

Reviewer 3 Report
Comments and Suggestions for Authors
The manuscript fits with the scope of Plants, and contribute with relevant data with respect to phytochemical content of acai, miriti and cupuassu. Quantitative investigations are welcomed, in particular with respect to species that have gained a market position within health applications.
That said, the present version of the manuscript doesn't point out what is new contribution to this science. This should be highlighted and revealed in the abstract. If there is no news, the manuscript makes no contribution but to confirm previous investigations! Which hypotheses were made prior to the investigation? On which basis did you start this investigation? I understand that you wanted to confirm the content of phenolics from the Vaupes-area. Make this clearer!
As far as I understand, there is no report on new chemical structures from these plant species? Neither is there any new methodologies in this work? Please explain and highlight!
The introduction should be clear and concise, and include a brief background of the science, a description of present holes in knowledge, and a rationale of the present contribution - which holes of knowledge do you intend to fill! As by now, there are much information about the species and their phytochemical contents. In addition, much is said about bioactivity of flavonoids. I would rather like a compilation of the present knowledge of phenolic content of the species, which would be in line with the present work. As you used lc-ms3q as the main analytical method, I would also like to see some information about this in the introduction section. Why lcms3q? This is still a cutting-edge method!
Methods: Which standards were used for quantification? Specify! What is meant by relative quantification?
Describe in more details the freeze-drying process?
I am a bit confused about the extraction procedures. It seems that there were made two extracts of each of the samples, one alcohol extract and one hydrolysis extract. Correct? Which extracts were analysed? Was the first extract analysed for its content of flavonoids (table 1) and the second for its content of phenolic acids (table 2)?
How many replicates were made for each sample? Analytical replicates or plant extract replicates?
Results and discussions: Highlight first of all what are the news in the present work!! Remember to make a discussion of your own data rather than discuss something else. I like the kind of discussion made in lines 154-158, and line 192 forward.
Table 1 and 2: These are important elements of the manuscript. As this is a chromatography work, I would be happy to see the tables content organized in accordance to increasing retention factors. This could then be connected to chromatograms as separate figures.
I do not see the meaning of the last column of the table (mean values of all species)? I would recommend to remove this column.
When you describe the quantitative content of the species, please include DM to let the reader know that you talk about dry matter content.
There is an important question left: Where are all the anthocyanins gone? Acai is a well-known source of anthocyanins! And the content of anthocyanins has been mentioned in the text already. Please explain and include!
Author Response
"Please see the attachment"

Round 2
Reviewer 1 Report
Comments and Suggestions for Authors
The authors have revised and expanded the original text. I consider the manuscript accepted, but some additional revisions must be made:
Introduction
Lines 67, 78, 88 and in other lines:
Botanical family names are not written in italics. The authors must correct this item throughout the text.
Lines 107-108:
The sentence was not clear: “Additionally, we investigate the flavonoid (anthocyanin) composition of these fruits using untargeted metabolomic approaches via …enabling the detection of a wide range of secondary metabolites.”
The authors, at the beginning of the sentence, report that they intended to analyze flavonoids, using untargeted metabolomic approaches? The objective of analyzing flavonoids seems to me to be a targeted analysis.
Lines 146 and 149: Please review the word açaí.
Line 148: In the nomenclature of glycosylated flavonoids, the oxygen atom that links the aglycone and the sugar must be written in italics. The authors should correct this item throughout the text.
Example: cyanidin-O-galactoside, kaempferol-O-glucoside
Results
Tables 1 and 2.
Authors should review the use of commas instead of periods in the results described in these tables.
I suggest that the authors carefully read the text for other minor revisions that should be made.
Author Response
"Please see the attachment"

Reviewer 3 Report
Comments and Suggestions for Authors
A few comments to this revision:
Table 1: Catehin is listed twice.
Line 172: Catechin and epicatechin are not flavonols. Call them catechins.
Chapter 3.5: Anthocyanins are rarely analysed by use of gas chromatography. Due to high hydrophilicity and low yields upon silylation, liquid chromatography is most frequently used. The authors should make a rationale for this choice of method. Furthermore, cyanidin 3-galactoside has not previously been reported to be a main constituent in acai, or?
Line 324: Sigma-Aldrich rather than Sygma -Aldrich.
Line 325: Hours rather than houts.
Author Response
"Please see the attachment"
